# Trends and Patterns in Electronic Health Record Research (1991–2022): A Bibliometric Analysis of Australian Literature

**DOI:** 10.3390/ijerph21030361

**Published:** 2024-03-19

**Authors:** Hongmei Xie, Andreas Cebulla, Peivand Bastani, Madhan Balasubramanian

**Affiliations:** 1College of Business Government and Law, Flinders University, Adelaide, SA 5042, Australia; peivand.bastani@flinders.edu.au (P.B.); madhan.balasubramanian@flinders.edu.au (M.B.); 2Australian Industrial Transformation Institute, College of Business Government and Law, Flinders University, Adelaide, SA 5042, Australia; andreas.cebulla@flinders.edu.au; 3Menzies Centre for Health Policy and Economics, School of Public Health, Faculty of Medicine and Health, The University of Sydney, Camperdown, NSW 2050, Australia

**Keywords:** electronic health record, electronic medical record, Australia, bibliometric analysis

## Abstract

Electronic Heath Records (EHRs) play vital roles in facilitating streamlined service provision and governance across the Australian health system. Given the recent challenges due to the COVID-19 pandemic, an ageing population, health workforce silos, and growing inefficiencies in traditional systems, a detailed historical analysis of the use of EHR research in Australia is necessary. The aim of this study is to examine the trends and patterns in EHR research in Australia over the past three decades by employing bibliometric methods. A total of 951 articles published in 443 sources were included in the bibliometric analysis. The annual growth rate of EHR research in Australia was about 17.1%. Since 2022, the main trending topics in EHR research were COVID-19, opioid usage, and natural language processing. A thematic analysis indicated aged care, clinical decision support systems, cardiovascular disease, drug allergy, and adverse drug reaction as the “hot” themes in EHR research in Australia. This study reveals a significant uptrend in EHR research in Australia, highlighting the evolving intellectual and collaborative landscape of this interdisciplinary field. The data also provide guidance for policymakers and funding institutions in terms of the most significant contributions and key fields of research while also holding public interest.

## 1. Introduction

The COVID-19 pandemic has substantially accelerated the adoption of healthcare technology, including the use of electronic health records (EHRs) in health and aged care systems in Australia [1,2,3,4,5,6]. In general, an EHR system is an integrated aggregate of patients’ health records developed and maintained by a variety of different healthcare organizations and kept in an electronic format [7]. EHRs contribute towards improving patient safety, raising the quality of care, increasing patient monitoring, and lowering healthcare costs. Despite the fact that EHR usage in many parts of the world has encountered several challenges in terms of interoperability, privacy, and ethical issues, EHRs play a significant role in meeting the ever-changing and demanding healthcare needs in the growing digital health ecosystem [8,9].

Over the past three decades, EHRs have evolved from transaction-focused systems to more intelligence-powered patient-centric systems [10]. Particularly in the past ten years, we have been seeing a rapid shift in the dominant design in EHR systems globally, which is mainly due to maturity and the successful implementation of Artificial Intelligence (AI) in the medical arena, as well as the availability of Big Data [1,11,12,13,14]. Patient engagement, interoperability, and predictive analytics have become key aspects in EHR system design. Through EHRs, patients can not only receive information such as reminders, alerts, and test results, but they can also share information and interact with health professionals through health monitoring devices or virtual technologies [2,15,16,17,18]. Interoperability is another area that has been important in patient-centred EHR design [19,20]. Previously, EHRs were usually created and maintained by a single provider. As a consequence of this, medical records have traditionally been kept in compartmentalized locations and computer databases; this presented challenges to the process of information sharing within a patient-centred integrated healthcare system [7,21]. Today, with the advent of Fast Healthcare Interoperability Resources (FHIR) and other consortia like Argonaut and the CARIN Alliance, the door is open to a better coordinated system through standardized Application Programming Interfaces (APIs) [22,23,24]. Predictive analytics and clinical decision support systems have also been able to draw actionable insights from the EHR system to inform our health services and individual health outcomes [25,26,27].

While EHR research has been undergoing an uptrend globally, there are few studies (mainly reviews or commentaries) that have systematically examined the growth and development of EHR research in Australia. EHR research in Australia has been reported in the literature for a variety of age groups, including children, adolescents, and older people; for a variety of health conditions, including cancer and heart diseases; and in a variety of care settings, including intensive care units and aged care facilities. The growth in EHR research in Australia requires considerable study to both understand the trends in key areas of research as well as to note hot topics and niche research areas. Consequently, the aim of this paper is to illustrate the trends in EHR research in Australia through a bibliometric analysis of the scientific literature. This paper will examine the trends and patterns in scientific outputs of EHR research and investigate frontiers in the context of Australia using publications from 1991 to 2022. This study provides insights into the EHR research landscape in Australia and will also help policymakers identify areas of EHR focus that are necessary for informed policy decisions in healthcare.

## 2. Materials and Methods

This study employs bibliometric methods to investigate the trends and patterns in Australian literature on electronic health records (EHRs) by conducting both a descriptive analysis and “performance analysis” [28,29] on the publications and using “science mapping” [30,31] techniques with knowledge structures. The analysis also provides a visualization of the data to assist in the identification of developing trends and hot topics. The purpose of a bibliometric analysis is to examine the multiple facets of a body of published literature using both quantitative and visual methods [32,33,34]. Bibliometric methods enable researchers to draw their conclusions on accumulated bibliographic data generated by other professionals in the field who express their ideas through citation, collaboration, and writing [35]. This method is also distinguished by its potential to play an essential part in the formulation of recommendations for public health, notably establishing guidelines or policies [36].

### 2.1. Search Strategy

A comprehensive systematic search was conducted utilising the Web of Science Core Collection (WoSCC) database, which was accessed through the Flinders University library. Web of Science is the world’s largest and most comprehensive academic information repository, encompassing the widest range of topics and fields possible, including the majority of the essential academic journals published in the field of EHRs [35,37]. The search was conducted and refined between November and December 2022. The data for this analysis were limited to the end of December 2022 using the WoSCC category with topic search (TS) shown in Box 1.

Box 1The bibliometric search strategy.
Concept 1: (electronic NEAR/2 Record*)Concept 2: (“australia*” OR “new south wales” OR “queensland*” OR “victoria” OR “south australia*” OR “tasmania*” OR “northern territory” OR “australian capital territory” OR “western australia” OR “brisbane” OR “sydney” OR “melbourne” OR “hobart” OR “adelaide” OR “darwin” OR “perth”)Concept #1 AND Concept #2


In Concept 1, the keywords were combined using “near/2” proximity operators to cover the commonly seen concepts “electronic health record*” OR “electronic medical record*” for the broadest and most relevant results. The search strategy also included using the Boolean operator “AND” to filter the results to the Australian context.

### 2.2. Data Processing

The initial search yielded 973 documents. To enhance the relevance of the analysed papers, specific selection criteria were implemented. Papers classified as letter, meeting abstract, editorial material, and news item were excluded, except when they met the classification criteria for article, early access, proceedings paper, or review article. In addition, papers published in languages other than English were excluded from the analysis. Therefore, 19 records were excluded from further analysis based on the above selection criteria. The remaining 954 documents were saved to the WoS Marked List. These documents were then exported to an Excel (version Excel for Microsoft 365) file for duplicate screening. Three duplicates were identified manually and removed from the collection. Sequentially, the WoS Marked List was updated with the final 951 documents as a source of truth for this EHR bibliometric study. To ensure transparency and accuracy, a PRISMA-style flowchart [38] was employed to outline the steps taken during the data gathering process (see Appendix A, Appendix A), and two team members conducted the process independently.

### 2.3. Data Analysis and Visualisation Tools

These 951 documents from the Marked List of the data source were exported as a plain text file (available upon request) with the selection of the full records and cited references for analysis in R (version 4.2.2) and R Studio (version 2022.12.0). In this study, the analysis conducted through bibliometrix library in R and biblioshiny web interface provided a comprehensive view of the scientific landscape, revealing the key trends, themes, and influential individuals and organizations in terms of EHR research over the past three decades in Australia. Our selection of a 31-year time period was based on the beginning of EHR research in Australia as documented in the WoSCC database in the early 1990s.

Bibliometrix is an R-tool used for comprehensive science mapping analysis, while biblioshiny is a shiny-based web app for user-friendly bibliometric analysis [39]. The analysis conducted in biblioshiny involved four levels of domain analysis on top of the overview to the dataset, namely sources, authors, affiliations, and documents [39]. In this study, we particularly focus on a high-level overview for identifying trends and areas of prolific research. In addition to the four levels of domain analysis, the research team also conducted three structures of knowledge analysis through science mapping, including conceptual, intellectual, and social analyses. These analyses helped to identify the underlying themes, concepts, intellectual structures, and social networks that shape EHR research. By applying a clustering algorithm on the keyword network, it is possible to highlight the different themes of a given domain. Each cluster/theme can be represented on a particular plot known as a strategic or thematic map [40]. In a thematic map, Callon Centrality can be read as the importance of a theme in the entire research field, while Callon Density can be read as a measure of the theme’s development [41,42].

Throughout the bibliometric analysis, and after careful consideration, the team selected Author’s Keywords over Keywords Plus (the citation indexes’ coding used by Web of Science). The decision to use Author’s Keywords was based on the belief that they provided a more comprehensive and up-to-date representation of the themes than Keywords Plus [43], which can be influenced by unknown factors. A synonyms file was developed to capture repeated words/word meanings. The default parameters provided by the biblioshiny tool were employed for all bibliometric analysis, unless otherwise stated in the results.

## 3. Results

### 3.1. Descriptive Analysis of the Publications

Table 1 presents a detailed summary of the EHR research trends in Australia based on an analysis of 951 documents published between 1991 and 2022 in WoSCC. The data reveal that the annual growth rate of EHR research in Australia is 17.14%, indicating a steady increase in research activity in this field over time. The average age of the documents in the collection is 5.98 years, suggesting that the research represented in the collection is relatively current. The average number of citations per document, which is 12.17, demonstrates the significant impact of the research and the widespread influence of the findings in the field. The trends of EHR publication and their citations can also be visualized in Figure 1.

According to Appendix A (Appendix A) of the most relevant authors, affiliations, sources, and locally cited sources, Georgiou A stands out as the most prolific author with 27 published articles, and Westbrook JI and Yu P follow closely behind with 23 and 18 published articles, respectively. The figure also indicates that the University of Melbourne, the University of Sydney, and Monash University are the most prolific contributors to EHR research in Australia, with 218, 201, and 196 articles, respectively. These universities have emerged as the leading centres of EHR research and development in Australia, demonstrating their strength and depth of expertise in this area. Furthermore, this figure illustrates that the *International Journal of Medical Informatics* has the highest number of publications with 42 articles, followed by the *Internal Medicine Journal* with 35 articles, while *Emergency Medicine Australasia* and the *Health Information Management Journal* are also notable sources, having published 27 and 26 articles, respectively. *The Medical Journal of Australia* tops the list as the most locally cited source with 532 citations, followed closely by the *Journal of the American Medical Informatics Association* with 528 citations. The *International Journal of Medical Informatics* also received significant attention from local researchers with 412 citations. Other highly cited sources include the *New England Journal of Medicine*, *JAMA-Journal of the American Medical Association*, and *The Lancet* with 296, 263, and 257 citations, respectively.

Additionally, the collection is associated with 1898 Keywords Plus (ID) and 2434 Author’s Keywords (DE), indicating the topics and themes addressed by the research. On average, each document in the collection has 5.85 co-authors. The data also show the importance of international collaboration in EHR research, as 19.77% of the co-authorships are international. The data indicate that journal articles are the most prevalent document type in the collection. Specifically, out of the total 951 documents included, 791 were categorized as articles, while 36 were classified as articles and early access papers, and 10 were identified as articles and proceedings papers.

### 3.2. Trend Topics in EHR Research in Australia Based on Author’s Keywords

Table 2 provides information on the frequency and temporal distribution of Author’s Keywords (DE) in the exported data related to electronic health record research in Australia. The most frequently occurring Author’s Keywords in the data are “electronic health records” (168), “electronic medical records” (69), “e-health” (37), “general practice” (36), and “primary care” (30). Other notable Author’s Keywords include “epidemiology” (28), “emergency department” (24), “hospitals” (22), “medical records” (21), and “residential aged care” (20). This table also includes keywords related to specific health conditions, such as cardiovascular disease (5) and mental health (13), as well as keywords related to implementation (13), evaluation (9), and education (12). Interestingly, there are five instances of the keyword “natural language processing,” indicating that this is a relevant new area of EHR research in Australia in 2022. Furthermore, “digital health” (8), “COVID-19” (14), and “opioid” (5) are also among the frequently used keywords in recent years. These Author’s Keywords can be visualized in the treemap format (see Appendix A, Appendix A) to provide a comprehensive understanding of their relative frequencies within the dataset as well as the duration of these topics.

### 3.3. Conceptual Structure

Figure 2 displays the clustering analysis results based on the values of the Callon Centrality and Callon Density measures for various EHR research themes in Australia. Callon Centrality is a measure of the strength of the connection between a given theme and the other themes in the dataset, while Callon Density measures the degree to which the themes in the cluster are interconnected. The cut-off values for Callon Centrality and Callon Density are determined by the median values of these measures across all clusters. In these EHR research data, the median Callon Centrality is 0.0625, while the median Callon Density is 22.

The identified clusters are subsequently categorized into one of the four quadrants based on their Callon Density and Callon Centrality scores. The upper right quadrant is indicative of highly dense and central clusters, which are labelled as “Motor Themes” in this study. The upper left quadrant comprises highly dense but less central clusters, referred to as “Niche Themes”. Similarly, the lower right quadrant represents highly central but less dense clusters, labelled as “Basic Themes”. Lastly, the lower left quadrant represents less central and less dense clusters, referred to as “Emerging or Declining Themes”. As shown in Figure 3, the “Motor Themes” quadrant includes clusters such as “residential aged care”, “adverse drug reaction”, “obesity”, “cardiovascular disease”, and “children”. The “Niche Themes” quadrant includes clusters such as “monitoring”, “HIV”, “ambulance”, “malnutrition”, “critical care”, and “rehabilitation”. The “Basic Themes” quadrant encompasses clusters such as “electronic health records”, “epidemiology”, “pregnancy”, “general practice”, “medical informatics”, and “mental health”. Finally, the “Emerging or Declining Themes” quadrant includes clusters such as “outcomes”, “pediatric”, “advance care planning”, “trauma”, and “health services”.

Figure 3 of thematic evolution provides further information in helping us understand the evolution of Australian EHR research across four time periods. One notable trend is the proliferation of terms and concepts within the field, indicative of a growing complexity and depth of investigation. Moreover, research efforts have expanded beyond the initial stage of exploration between the 1990s and 2010, with new terms and concepts emerging, including “big data”.

### 3.4. Intellectual Structure

In this study, we explored intellectual structure using the Historiographic Mapping technique. This approach helps to identify historical paths for each research topic, as well as the core authors and documents associated with the topic [44]. Figure 4 displays the historical evolution of the 30 most cited works in the field of EHRs from 1991 to 2022 in Australia. The nodes in the graph represent individual works labelled with the first author and the year of publication. Each edge represents a direct citation, with the arrows indicating the direction of the citation. The size of each node corresponds to the number of local citations (LCS) that the work has received, while the colour of the node reflects the cluster to which the document belongs. The graph is segmented into four distinct clusters, which represent different research topics and subfields within the EHR domain.

One of the most prominent clusters in red in the graph centres around the work of Keith McInnes, whose 2006 article, “General Practitioners’ Use of Computers for Prescribing and Electronic Health Records: Results from A National Survey”, is one of the most cited works in the EHR field. This cluster includes eight documents, which is the second biggest cluster of this graph.

Another significant cluster in blue is the biggest cluster including 15 documents. In this cluster, the most cited works are those by Deutsch (2010), titled “Critical Areas of National Electronic Health Record Programs-Is Our Focus Correct?”; Zhang (2012), titled “The Benefits of Introducing Electronic Health Records in Residential Aged Care Facilities: A Multiple Case Study”; and Andrews (2014), titled “The Australian General Public’s Perceptions of Having a Personally Controlled Electronic Health Record (Pcehr)”. They each have eight local citations.

A third cluster in green contains two documents. In this cluster, Hawley’s work (2014), titled “Sharing of Clinical Data in a Maternity Setting: How Do Paper Hand-Held Records and Electronic Health Records Compare for Completeness?” cited O’sullivan (2011), titled “Just What the Doctor Ordered: Moving Forward with Electronic Health Records”.

A fourth cluster in purple contains two documents. In this cluster, Bernardo’s work (2019), titled “Influenza Immunization Coverage from 2015 to 2017: A National Study of Adult Patients from Australian General Practice”, cited Gonzalez-Chica DA’s 2018 article.

Overall, the graph serves as a visual representation of the EHR field’s historical development, highlighting the key works and research topics that have shaped the field over the past three decades.

### 3.5. Social Structure

In this study, we use “institutional collaboration” with Betweenness Centrality [45] and PageRank [46] to explore social structure. Figure 5 provides information on the collaboration network in electronic health record (EHR) research in Australia. The nodes represent institutions involved in EHR research, while the colours indicate their clustering. There are two major clusters in this network, namely one in red and another in brown.

The red cluster in the network reveals a high level of collaboration and interconnectedness among the institutions. Within this cluster, the University of Sydney had the highest Betweenness Centrality score (181.367522356292) and a high PageRank score (0.0707735905254743), indicating its importance in the network. Other institutions within this cluster, including the University of New South Wales, the University of Wollongong, Macquarie University, and Deakin University, also had relatively high Betweenness and PageRank scores. Meanwhile, the brown cluster, represented by the University of Melbourne, had the highest PageRank score (0.080113721), suggesting its significant influence in the network.

## 4. Discussion

This study explored the landscape of electronic health record (EHR) research in Australia from 1991 to 2022 through the integration of two bibliometric analysis techniques. The first approach involved a “performance analysis”, which primarily entailed a descriptive examination of the publications. The second approach, “science mapping”, employed knowledge structures to uncover latent patterns in the data. Through the combination of these methods, a comprehensive understanding of the publication trends, the intellectual and social structures, and key themes of EHR research in Australia was achieved.

EHR-related research in Australia has shown a growing trend over the past 31 years, with a significant increase in the past decade. The top five prolific institutions have been concentrated in economically developed provincial states. This is consistent with global trends in EHR research [37]. The increasing volume of studies and the evolving nature of EHR technologies can inform policy decisions aimed at integrating cutting-edge EHR solutions into clinical practice, improving patient care, and enhancing healthcare efficiency. Moreover, the significant incremental trend in research activity underscores the readiness of the Australian healthcare sector to embrace digital transformations, positioning it to leverage EHR advancements in addressing contemporary healthcare challenges, such as data interoperability, patient privacy concerns, and the need for real-time data analytics in clinical decision making. Therefore, it becomes imperative for policymakers to engage with these research findings actively, translating academic insights into practical strategies that capitalize on the potential of EHR systems to revolutionize healthcare delivery.

The thematic map of EHR research studies at a global level identified themes, such as “clinical decision support”, “primary care”, “epidemiology”, “diabetes”, “palliative care”, “cancer”, “COVID”, and “telehealth” [37], which, by and large, align with the present study in the Australian context. Similar themes have been observed in other studies, such as in research conducted in Brazil [47] and in Europe [48,49,50], which underscores the universal drive towards interoperability, advancements in cloud computing and security, and the integration of AI and analytics for better decision making and patient care. These commonalities suggest a global consensus on the prioritized areas within EHR research.

However, our study also highlights the growing importance of aged care as a prominent EHR-related theme. The “Motor Themes” quadrant, which includes clusters such as “residential aged care”, “clinical decision support”, “obesity” and “cardiovascular disease”, represents highly dense and central research areas in the Australian EHR landscape, indicating continued research interest and innovation in these areas. The observed trends may be attributed to two factors. Firstly, the Australian government’s response to the findings of the Royal Commission into Aged Care Quality and Safety has prioritized the reform of the elderly care sector, resulting in increased funding towards this area [51,52]. Secondly, the COVID-19 pandemic has accelerated the adoption of technology, including the use of EHRs, in healthcare [1,2,3,4,6]. These factors highlight the need for continued research and innovation in EHRs to address the growing healthcare needs of Australia’s aging population.

This study not only maps the landscape of EHR research, but also pinpoints areas that need practical application and further investigation. For instance, the surge in EHR studies related to COVID-19 underscores the system’s role in pandemic response, facilitating real-time data sharing which is crucial for public health decisions. This finding suggests that future research could delve into optimizing EHR systems for better pandemic preparedness and response, potentially through developing advanced predictive analytics tools within EHR platforms.

Additionally, the emphasis on aged care within our analysis highlights an opportunity to leverage EHR technologies for improving medication management among the elderly, specifically through EHR-based alerts for drug interactions and the potential integration of telehealth service. This application directly addresses the challenges of managing complex medication regimes, illustrating a clear pathway from bibliometric insights to healthcare practice improvements. In consideration of these outcomes, it becomes crucial for policymakers and healthcare providers to strategize the advancement of EHR systems within the Australian setting by fostering user acceptance and integrating EHRs with the aged care system’s monitoring and surveillance frameworks, as well as with disease registries. The implication of a disease registry is also evident in a Brazilian study [53]. These efforts will not only elevate the standard of elderly care, but also tackle the issues posed by an aging population and the ramifications of the COVID-19 pandemic.

However, the findings of this study should be interpreted in the context of a few limitations. First, it has been argued that Scopus provides a more comprehensive coverage of sources than the Web of Science (WoS) database in fields beyond medicine and physical sciences [54], while WoS is generally regarded as the source with the highest quality of information. Consequently, there is a possibility that some important papers may have been missed. To overcome this limitation, one potential solution is to utilize additional sources, such as Scopus; compare the differences in records; and incorporate any missing papers in future studies. Second, despite the fact that a bibliometric methodology leverages the precision of quantitative methods, it also encourages reviewers to apply field knowledge toward the synthesis and interpretation of results emerging from the thematic analysis. The validation of identified themes/subdomains is crucial in using bibliometric mapping as a policy-supporting tool, and it requires the input of field experts to confirm the identified themes [30]. To facilitate the confirmation of themes by field experts, it would be beneficial to further explore the use of an alluvial graph in conjunction with Publication Year Spectroscopy (RPYS). RPYS can trace the historical origins of research fields/topics [55], while an alluvial graph in a thematic map, which divides the time span into time slices, can track temporal changes over time. Thus, these tools can be useful to reveal the evolution of research themes in a specific field of EHRs. Third, a critical limitation of our study is the constrained exploration of ethical and safety considerations associated with EHRs. Given the substantial importance of ethical issues—ranging from patient privacy and data security to the ethical use of EHR data for research and clinical decision making—our analysis acknowledges the need for a deeper investigation in this area. The ethical landscape surrounding EHRs is complex and evolving, requiring thorough consideration beyond the scope of our current bibliometric methodology. Future research should aim to specifically address these ethical and safety concerns, incorporating qualitative analyses and expert consultations to comprehensively explore the implications of EHR use on patient care and privacy. The scope of the bibliometric search is limited to Australian works. Based on this, although health policymakers in settings similar to Australia may find these results appropriate, it is suggested to interpret the findings conservatively and tailor the results according to local settings.

## 5. Conclusions

This study identified a steady increase in EHR-related research through the years, indicating the growing interest in the field. A strong level of collaboration was observed among authors and institutions, which highlights the importance of inter-disciplinary research in this area. This comprehensive descriptive analysis, together with the use of the Science Mapping technique, presents a synthesized overview of the topic in this specialized area of EHR in general and for the Australian context in particular. Similar to the global trends and related themes on the priority areas within EHR research, the Australian context shows the same priorities in the areas of clinical decision support systems, cardiovascular diseases, drug allergies, and adverse drug reactions. Overall, this study offers insights into the knowledge structures of EHR-related research and could serve as a valuable reference point for scholars and policymakers alike. The identification of experts and prominent institutions in this field could facilitate informed policy discussions and funding allocations, ultimately benefiting healthcare practices and patient outcomes.

## Figures and Tables

**Figure 1 ijerph-21-00361-f001:**
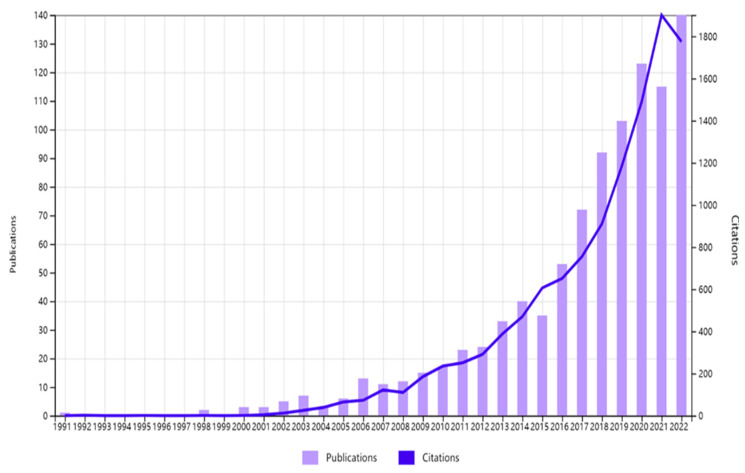
Trends in publications and citations from 1991 to 2022.

**Figure 2 ijerph-21-00361-f002:**
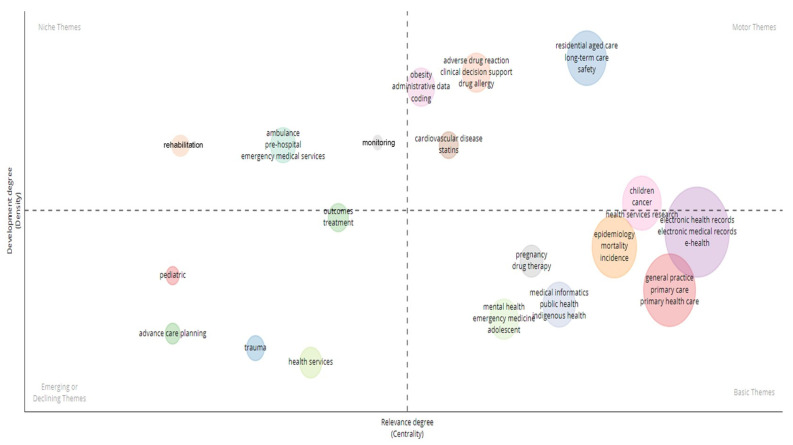
Conceptual structure—thematic map.

**Figure 3 ijerph-21-00361-f003:**
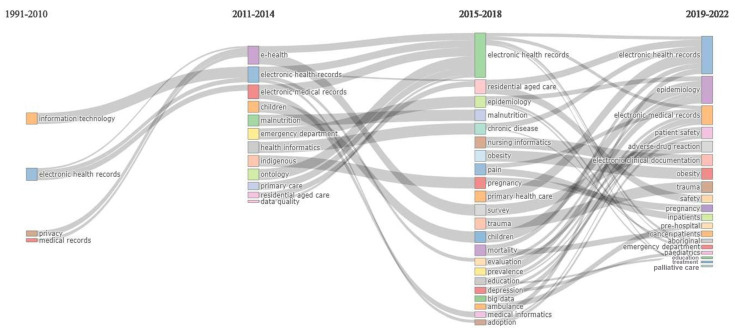
Thematic evolution of EHR research in Australia across four time periods.

**Figure 4 ijerph-21-00361-f004:**
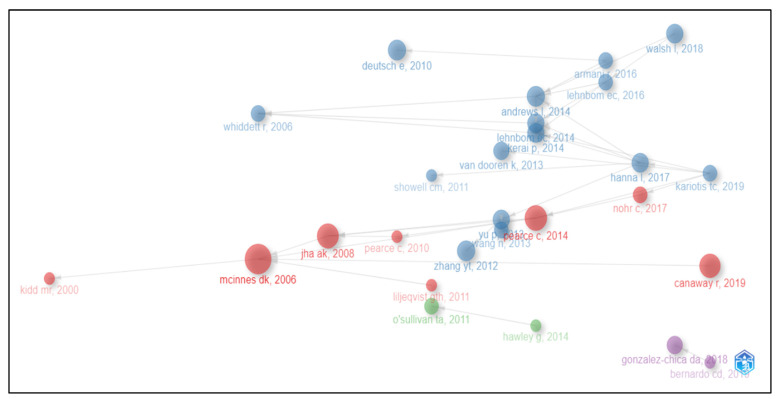
Intellectual structure. Bibliometric historiography showing historical evolution of the most cited works.

**Figure 5 ijerph-21-00361-f005:**
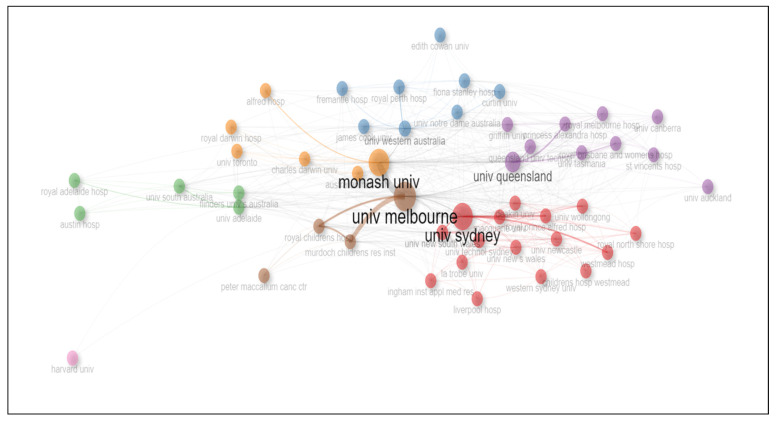
Social structure—collaboration network based on affiliations.

**Table 1 ijerph-21-00361-t001:** Summary characteristics of included studies.

Description	Results
Main information about data	
Timespan	1991:2022
Sources (journals, books, etc.)	443
Documents	951
Annual growth rate %	17.14
Document’s average age	5.98
Average citations per doc	12.17
References	25,752
Document contents	
Keywords Plus (ID)	1898
Author’s Keywords (DE)	2434
Authors	
Authors	4289
Authors of single-authored docs	28
Authors collaboration	
Single-authored docs	30
Co-authors per doc	5.85
International co-authorships %	19.77
Document types	
article	791
article; early access	36
article; proceedings paper	10
editorial material; early access	1
proceedings paper	76
review	36
review; early access	1

**Table 2 ijerph-21-00361-t002:** Trend topics in EHR research in Australia.

Item	Freq	Year_q1	Year_med	Year_q3
electronic health records	168	2014	2017	2020
electronic medical records	69	2015	2019	2021
e-health	37	2014	2016	2019
general practice	36	2019	2020	2021
primary care	30	2017	2019	2020
epidemiology	28	2017	2019	2021
emergency department	24	2017	2020	2021
hospitals	22	2018	2020	2021
medical records	21	2013	2017	2018
residential aged care	20	2015	2018	2020
privacy	14	2011	2014	2017
COVID-19	14	2021	2022	2022
implementation	13	2011	2018	2021
mental health	13	2017	2018	2020
education	12	2015	2017	2018
public health	12	2018	2021	2021
information management	11	2008	2015	2019
health information systems	10	2013	2016	2017
inpatients	10	2017	2021	2021
evaluation	9	2013	2015	2020
personal health records	9	2015	2016	2017
digital health	8	2020	2021	2022
audit	7	2011	2013	2018
data quality	6	2014	2014	2016
confidentiality	6	2012	2015	2017
information systems	5	2010	2011	2017
quality	5	2013	2013	2019
cardiovascular disease	5	2013	2014	2018
natural language processing	5	2022	2022	2022
opioid	5	2021	2022	2022

## Data Availability

The data presented in this study are available upon request from the corresponding author.

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
