# Peer review of "Trends and Patterns in Electronic Health Record Research (1991–2022): A Bibliometric Analysis of Australian Literature"

_ijerph, 2024, doi:10.3390/ijerph21030361_

Round 1
Reviewer 1 Report
Comments and Suggestions for Authors
This article provides a comprehensive and timely analysis of the trends and patterns in EHR research in Australia, shedding light on the evolving landscape of this critical field.
1. The paper would benefit from a more detailed discussion of the practical implications of the identified "hot" themes in EHR research in Australia, including how these themes can inform future research, policy decisions, and healthcare practices.
2. EHR research is a huge domain. It covers various issues/ factors/ challenges related to Interoperability, security, privacy, standardization, data quality, breaches etc to name a few. The paper must classify the reserach domain focus on some of the parameters despite a general approach.
3. It would be valuable for the authors to provide a more in-depth analysis of the potential impact of the uptrend in EHR research in Australia on addressing the specific challenges mentioned, such as the COVID-19 pandemic, ageing population, and inefficiencies in traditional systems, to provide a clearer link between the research findings and real-world healthcare improvements.
4. The authors should consider providing a discussion on the ethical considerations and implications of the study, particularly regarding data privacy and security in the context of EHR research, to address the ethical dimensions of the research topic.
5. The article could be strengthened by providing specific examples of how the insights from the study could be practically applied to drive meaningful advancements in Australian healthcare practices and patient outcomes.
6. The quality of all the figures(1,2,3,4) needs to be improved.
7. More references from 2023 needs to be covered.
Comments on the Quality of English LanguageSome grammar checks are required.
Author Response
Please consider our detailed and point-by-point answers to reviewer one`s comments attached.

Reviewer 2 Report
Comments and Suggestions for Authors
The manuscript "Trends and Patterns in Electronic Health Record (EHR) Research (1991-2022): A Bibliometric Analysis of Australian Literature" addresses a relevant and really important topic at this time of digital transformation in global health.
The recommendations made are aimed at improving the final quality of the article.
Recommendations
[1. Introduction
The text is very good, but it would be important for the authors to include data from other countries or regions of the world, so that readers can make comparisons. As the manuscript is centered on Australia, it is difficult to have a reference that allows us to truly understand the level of technological maturity in this country.
[2] Figures
All figures and graphics used need to improve quality (resolution), as they appear distorted in the manuscript, making them unreadable. Authors should pay attention to the recommendations of this scientific journal regarding the quality of the images used.
[3] Discussion
The authors present in the discussion some themes that were found at a global level.
"The thematic map of EHR research study at a global level identified the themes such 311 as “clinical decision support”, “primary care”, “epidemiology”, “diabetes”, “palliative 312 care”, “cancer”, “COVID” , and “telehealth” [37], which by and large align with this present study for the Australian context"
However, the authors could go deeper into these issues, for example, highlighting in which countries or regions these discussions are taking place. In Brazil, for example, there is a broad discussion about the digital transformation of health and it covers exactly this topic. The manuscript "Electronic health records in Brazil: Prospects and technological challenges" addresses these issues, in addition to another study that addresses the use of Digital Health Solution for Monitoring and Surveillance of Amyotrophic Lateral Sclerosis.
The authors highlight the concern of the elderly population, however it would be interesting to compare with other countries, with the aim of presenting a comparative reference.
Author Response
Please consider our detailed and point-by-point answers to reviewer two`s comments attached.

Reviewer 3 Report
Comments and Suggestions for Authors
The use of EHR brings numerous benefits to clinical practice although it is not without problems. Although we have witnessed the technological development of EHR in recent decades, there are still gaps and points that must be improved to achieve the promised potential of this technology in clinical practice. To identify strengths and weaknesses of EHR research as well as discover trends and gaps, works such as the one proposed by the authors in this article are of great interest.
The article is well written and structured. However, there are several points that could be improved to strengthen the interest of the work for a wide audience:
1. The scope of the bibliometric search is limited to Australian works, and the results are not translated to other countries. The interest is thus restricted only to Australian readers (or those interested in the Australian setting). It would be interesting to include some literature reviews on EHR research in other countries or generally, and connect the findings from Australia to other publications from other countries. Does Australia have different research trends than other countries or is it aligned with global EHR research?
2. Although the Discussion indicates that the use of WoS may have limited the results, it would be necessary verify that using Scopus does not increase the number of relevant articles. This reviewer has done the Box 1 search for keywords in Scopus and has obtained 1160 articles. The exclusive use of WoS can lead to a significant bias in the results.
3. The discussion is interesting but should be improved. The interesting thing about this kind of works is conclusions drawn from bibliometric analyses and, in this case, the conclusions are few and once again, not compared to the global context.
4. Some bibliometric data is not as interesting as others and could be eliminated. For example, section 3.1 includes information about the most relevant authors and their affiliation, and what is interesting is the relevance of the articles (by citations) but not who published them.
5. Table 3 and Figure 2 are redundant. The table could be deleted.
6. The following figures are not displayed correctly: 1, 2, 3, 4, S2
7. It is not explained where Figure S4 comes from
8. Line 233: Figure 3 > Figure 2
Author Response
Please consider our detailed and point-by-point answers to reviewer three`s comments attached.

Round 2
Reviewer 3 Report
Comments and Suggestions for Authors
The authors have reviewed the manuscript according to this reviewer's comments successfully. They have made appropriate changes and when they have not, they present their reasons coherently. This reviewer has no further comments.